C3 plant isotopic variability in a boreal mixed woodland: implications for bison and other herbivores

http://orcid.org/0000-0001-9450-512X Metcalfe Jessica Z. jmetcal1@lakeheadu.ca
Department of Anthropology, Lakehead University , Thunder Bay, Ontario , Canada
Potts Alastair
Electronic publication date: 2021 Sep 23
Publication date: 2021
Volume: 9
Electronic Location ID: e12167
Received 2021 Jun 14; Accepted 2021 Aug 26
Copyright: © 2021 Metcalfe
Copyright year: 2021
Copyright holder: Metcalfe
License: This is an open access article distributed under the terms of the Creative Commons Attribution License, which permits unrestricted use, distribution, reproduction and adaptation in any medium and for any purpose provided that it is properly attributed. For attribution, the original author(s), title, publication source (PeerJ) and either DOI or URL of the article must be cited.
License URL: https://creativecommons.org/licenses/by/4.0/

Keywords: Stable isotopes, Carbon isotope, Nitrogen isotope, Terrestrial, Plant, Bison, Boreal, Isotopic baseline, Mammoth, Grass

Funding: Social Sciences and Humanities Research Council of Canada (SSHRC) Banting Postdoctoral Fellowship This work was supported by a Social Sciences and Humanities Research Council of Canada (SSHRC) Banting Postdoctoral Fellowship. The funders had no role in study design, data collection and analysis, decision to publish, or preparation of the manuscript.

==============================
Plant isotopic baselines are critical for accurately reconstructing ancient diets and environments and for using stable isotopes to monitor ecosystem conservation. This study examines the stable carbon and nitrogen isotope compositions (δ13C, δ15N) of terrestrial C3 plants in Elk Island National Park (EINP), Alberta, Canada, with a focus on plants consumed by grazers. EINP is located in a boreal mixed woodland ecozone close to the transition area between historic wood and plains bison habitats, and is currently home to separate herds of wood and plains bison. For this study, 165 C3 plant samples (grasses, sedges, forbs, shrubs, and horsetail) were collected from three habitat types (open, closed, and wet) during two seasons (summer and fall). There were no statistically significant differences in the δ13C or δ15N values of grasses, sedges, shrubs and forbs. On the other hand, plant δ13C and δ15N values varied among habitats and plant parts, and the values increased from summer to fall. These results have several implications for interpreting herbivore tissue isotopic compositions: (1) consuming different proportions of grasses, sedges, shrubs, and forbs might not result in isotopic niche partitioning, (2) feeding in different microhabitats or selecting different parts of the same types of plants could result in isotopic niche partitioning, and (3) seasonal isotopic changes in herbivore tissues could reflect seasonal isotopic changes in dietary plants rather than (or in addition to) changes in animal diet or physiology. In addition, the positively skewed plant δ15N distributions highlight the need for researchers to carefully evaluate the characteristics of their distributions prior to reporting data (e.g., means, standard deviations) or applying statistical models (e.g., parametric tests that assume normality). Overall, this study reiterates the importance of accessing ecosystem-specific isotopic baselines for addressing research questions in archaeology, paleontology, and ecology.

Introduction

The significant difference between the stable carbon isotope compositions (δ13C) of C3 versus C4 plants is the foundation for many paleodiet, foodweb, and conservation studies. However, terrestrial plants utilizing C4 photosynthesis are rare in cool high-latitude environments, including most of Canada, Europe, and northern Asia (Lüttge, 2004; Osborne et al., 2014; Still et al., 2003). During cold intervals such as the Last Glacial Maximum, C3-dominated environments extended to even lower latitudes (Cotton et al., 2016). Despite the lack of C4 plants, animal isotopic niche partitioning can still occur within C3-dominated areas because of predictable variations in C3 plants in response to factors such as aridity, soil salinity, degree of canopy cover, carbon source (atmospheric or aquatic), nitrogen source, and mycorrhizal associations. For example, terrestrial herbivores across Pleistocene Eurasia and North America occupied different isotopic dietary niches which varied temporally and geographically (e.g., Bocherens, 2015; Bocherens et al., 2015; Fox-Dobbs, Leonard & Koch, 2008; Metcalfe, Longstaffe & Hodgins, 2013; Metcalfe et al., 2016; Schwartz-Narbonne et al., 2019). Isotopic niche partitioning has also been demonstrated among modern terrestrial herbivores in C3-dominated environments (e.g., Ben-David, Shochat & Adams, 2001; Cerling, Hart & Hart, 2004; Feranec, 2007; MacFadden & Higgins, 2004; Stewart et al., 2003; Urton & Hobson, 2005). Interpreting the underlying causes of animal niche partitioning requires an understanding of local baseline isotopic variations (Casey & Post, 2011).

Processes underlying variations in δ13C values of terrestrial plants utilizing the C3 photosynthetic pathway have been reviewed elsewhere and are described only briefly here. Terrestrial C3 plants have δ13C values ranging from about −37 to −20‰ when standardized to a atmospheric CO2 δ13C of −8.0‰ (Kohn, 2010). Environmental factors known to affect C3 plant δ13C values include the isotopic composition and concentration of utilized CO2, sources of CO2 (atmospheric vs. aquatic, ancient vs. modern), water availability and plant water-use efficiency, soil salinity, degree of canopy cover, and plant type/taxa (e.g., Hare et al., 2018; Lajtha & Michener, 1994; Tieszen, 1991). Different parts of the same plant (e.g., photosynthetic vs non-photosynthetic tissues) can have widely disparate δ13C values as a result of different formation times, biochemical compositions, fractionations during transportation of biomolecules within the plant, and height within the forest canopy (Cernusak et al., 2009; Chevillat et al., 2005; Ghashghaie & Badeck, 2014). Seasonal changes in plant δ13C can occur due to differing environmental conditions during growth and/or changes during maturation (e.g., Lowdon & Dyck, 1974; Vogado et al., 2020). Variable isotopic compositions at the base of the food chain can be passed on to herbivores with differential feeding strategies (Casey & Post, 2011). For example, caribou/reindeer tend to have high δ13C values relative to co-existing herbivores because of their reliance on high-13C lichen, and animals that feed in closed-canopy areas have lower δ13C values than those that feed in open areas (e.g., Barnett, 1994; Drucker et al., 2010).

Nitrogen isotopic variability in plants results from utilization of different molecular forms of nitrogen, manner of nitrogen uptake (e.g., particular mycorrhizal associations) location of nitrogen assimilation, and mobilization of nitrogen within the plant (Craine et al., 2009; Hobbie & Hogberg, 2012). Temperature, aridity, mycorrhizal type, and degree of nitrogen cycling within an ecosystem have been shown to affect plant δ15N (see Szpak, 2014 for review). Aquatic versus terrestrial growth can also systematically affect δ15N values (Plint, Longstaffe & Zazula, 2019). Individual plant δ15N can change over time due to a range of factors, including growth stage, seasonal conditions, soil nitrogen conditions, and decomposition (Karlsson, Eckstein & Weih, 2000; Szpak et al., 2012; Tahmasebi et al., 2017). Variations in nitrogen isotopic compositions at the base of the food chain can be passed on to consumers, leading to significant variability in δ15N even among animals feeding at the same trophic level (Casey & Post, 2011). For example, differences in the δ15N of various members of the beaver family (Castoridae) likely reflect differing reliance on aquatic versus terrestrial woody plants (Plint et al., 2020; Plint, Longstaffe & Zazula, 2019, and the high δ15N values of mammoths (Mammuthus spp.) can be attributed to selection of high-15N grasses (Bocherens, 2003; Metcalfe, Longstaffe & Hodgins, 2013; Schwartz-Narbonne et al., 2015).

Plant isotopic baselines for archaeological and ecological studies are crucial for interpreting the isotopic compositions of ancient humans and animals. Failure to understand or account for variations at the base of the food chain can lead to incorrect interpretations of diet, trophic level, and environmental conditions, particularly when comparing among regions or time periods (Casey & Post, 2011). However, obtaining appropriate plant isotopic baselines for a region or time period of interest can be difficult. Published surveys of modern plant natural isotopic variability are relatively rare, and the majority of those that do exist report only means, standard deviations, and data visualizations rather than a full list of the measured isotopic compositions of individual plants (Table 1). Furthermore, compilations of regional or global plant isotopic data could obscure systematic variations that occur on a local level (see discussion in Drucker et al., 2010). Thus, ecosystem-specific baselines are ideal. Ancient plants are rarely preserved except in rare depositional environments (dry caves, permafrost) or as charred remains of cooking activities (e.g., Metcalfe & Mead, 2019; Styring et al., 2013; Szpak & Chiou, 2019; Wooller et al., 2007), which means that archaeological and paleontological studies must rely at least in part on insights from modern plants. This is certainly true in boreal environments, where highly acidic soils often cause complete degradation of organic remains (Gordon & Buikstra, 1981; Woywitka, 2016).

Table 1 Studies of modern C3 plant natural isotopic variability in cold-temperate and boreal locations.

Environment (s)	Location	Plant life-forms	Isotopes	Full data?	Reference	
Alpine meadow and steppe	Tibetan plateau	Grass, sedge, shrub, sub-shrub, forb	N	No	(Zhou et al., 2016)	
Alpine meadow and steppe	Tibetan plateau	Grass, sedge, shrub, sub-shrub, forb	C	No	(Zhou et al., 2013)	
Arctic	Alaska (Central & North)	Sedge, shrub, forb, lichen, moss	N	No	(Nadelhoffer et al., 1996)	
Arctic/Boreal	Alaska (Interior)	Grass, sedge, horsetail, herb	C, N	No	(Funck et al., 2020)	
Arctic/Boreal	Alaska (north-central)	Shrub/tree, aquatic	C, N	No	(Kielland, 2001)	
Arctic/Boreal	Alaska & Yukon	Shrub/tree, herb, aquatic, fungus, lichen	C, N	No	(Ben-David, Shochat & Adams, 2001)	
Arctic/Boreal	Alaska & Yukon	Grass, sedge	C	Yes	(Wooller et al., 2007)	
Arctic/Boreal	Alaska (Seward Pen.)	Graminoid, shrub, forb, lichen	N	No	(Finstad & Kielland, 2011)	
Arctic/Taiga	Alaska (taiga)	Shrub/tree	C, N	No	(Kielland & Bryant, 1998)	
Arctic/Tundra	Alaska (North Slope)	Graminoid, forb, shrub, lichen	C, N	Yes	(Barnett, 1994)	
Arctic/Tundra	Banks Island, NWT	Shrub, forb, grass, sedge, lichen, moss	C, N	Yes	(Munizzi, 2017)	
Boreal forest	N. Saskatchewan	Tree, shrub, forb, moss, lichen	C	No	(Brooks et al., 1997)	
Boreal Grasslands	S. Yukon	Shrub, herb	C, N	Yes	(Tahmasebi et al., 2017)	
Cold-temperate/boreal forests	North America & Eurasia	Tree	C, N	No	(Kloeppel et al., 1998)	
Coniferous forests	Oregon	Tree	C	No	(Bowling et al., 2002)	
Grasslands, fens, Salix snowbeds	Greenland	Graminoid, shrub	C, N	No	(Kristensen et al., 2011)	
Plains Grasslands	South Dakota & Mongolia	Tree, shrub, forb, grass	C	No	(Tieszen, 1994)	
Shrubland, meadow, steppe, desert	Tibetan plateau	Graminoid, tree, shrubs, xeromorph	C	No	(Song et al., 2008)	
Steppe	Inner Mongolia	Grass, shrub, sub-shrub, forb	C	No	(Chen et al., 2005)	
Subarctic woodlands/shrublands	Sweden (north)	Graminoid, woody, cryptogam	N	No	(Karlsson, Eckstein & Weih, 2000)	
Temperate forest	Switzerland	Tree	C	No	(Chevillat et al., 2005)	
Temperate grasslands/woodlands	UK	Grass	C, N	No	(Bonafini et al., 2013)	
Temperate to sub-Mediterranean	Germany & France	Tree	C, O	No	(Keitel et al., 2006)	
Various	N. Scandinavia	Tree, shrub, dwarf shrub	C	No	(Loader & Rundgren, 2006)	
Various	Canada (south to north)	Grass, tree, fern	C	Yes	(Lowdon & Dyck, 1974)	
Wetlands	Yukon & S. Ontario	Tree, shrub, aquatic	C, N	Yes	(Plint, Longstaffe & Zazula, 2019)	

Boreal mixed woodlands are important regions for understanding animal ecology and human-animal interactions. In particular, the plains-parkland transition in northern Alberta (Canada) was a critical area for both human and animal migrations, beginning with the opening of the so-called Ice-Free Corridor and continuing throughout the Late Holocene (e.g., Heintzman et al., 2016; Ives, 2003; Shapiro et al., 2004). Northern Alberta is home to a diverse mammalian fauna including ungulates such as moose, elk, and deer. Until the late 19th century, the region was also home to abundant bison, and was an area of transition between wood bison (Bison bison athabascae) territory in the north (i.e., boreal forests of northern Alberta and Saskatchewan, the Northwest Territories, Yukon, and Alaska) and plains bison (Bison bison bison) territory in the south (i.e., the prairies and plains) (Van Zyll de Jong, 1986). The current research was motivated by a desire to use stable isotope analysis to better understand modern and archaeological/paleontological bison dietary selectivity in C3-dominated boreal regions, where bison have access to a range of plants and habitats. As a first step, this study examines natural variations in the carbon and nitrogen isotope compositions of plants in Elk Island National Park (EINP), Alberta, with a focus on plants that may have been consumed by bison.

Study location: Elk Island National Park, Alberta

Elk Island National Park (EINP) is a ~200 km2 protected area located ~40 km east of Edmonton. The park is located within Canada’s southern boreal plains ecozone, an area of transition between semi-arid prairie and wetter boreal forest (Nicholson, 1995). Topographically, the park is part of the Beaver Hills region, an area of knob-and-kettle terrain with abundant lakes and wetlands. Vegetation within the park is a patchy mosaic of aspen parkland, boreal mixed woodland, grassy/shrub meadows, marshes, and lacustrine areas (Fig. 1) (Best & Bork, 2004; Holsworth, 1960; Nicholson, 1995). All identified plant taxa in the park utilize C3 photosynthesis (Hanna Schoenberg, personal communication, May 18, 2021). EINP’s mean annual temperature was 1.7 °C and mean annual precipitation was 460 mm between 1951 and 1980, but both temperature and precipitation have been increasing due to climate change (https://climatedata.ca/). EINP typically experiences moderate summers and cold, dry, windy winters. Temperatures range from average lows of −18 °C in January to average highs of 23 °C in July (weather-atlas.com). Peak summer rains occur in July (mean of 112 mm precipitation) and snowfall reaches a high of 206 mm in March (weather-atlas.com). Spring blooms typically begin to appear in May and the growing season lasts from approximately mid-May to mid-September.

Figure 1 Location of Elk Island National Park and plant sampling locations relative to vegetation zones defined in a previous Parks Canada survey.

During our sample collection, P1 and P3 were open and dry (not wet), whereas P7 was a small wetland (not shrubland). Other vegetation zones for sampling locations agreed with field observations during sampling. Map credit: Parks Canada, OGL.

EINP is home to several large ungulate species, including moose (Alces alces), elk/wapiti (Cervus canadensis), white-tailed deer (Odocoileus virginianus), mule deer (Odocoileus hemionus), plains bison (Bison bison bison), and wood bison (Bison bison athabascae) (Telfer & Cairns, 1986). For many decades, EINP has been a source for genetically-pure disease-free bison that have been introduced to conservation herds across the continent (Markewicz, 2017). The plains and wood bison areas are separate: plains bison range freely within the fully-fenced northern portion of the park and wood bison range freely within the separate, fully-fenced southern portion of the park (Fig. 1). Bison in both areas have access to the same types of habitat and vegetation.

Materials & Methods

Sample collection and preparation

Plant samples were collected with the permission of Parks Canada (Research and Collection Permit EI-2016-21863). Grass, sedge, forb, shrub, and horsetail (Equisetum spp.) samples were collected on June 27–29, 2016 (n = 133) and November 6, 2016 (n = 32) from dry open areas (e.g., meadows, hill slopes), dry closed-canopy environments, and wet areas (shorelines of ponds or lakes) (Figs. 1, 2). Site categorizations were based on observations at the time of sampling rather than on generalized vegetation maps, because wetlands can be ephemeral. Sampling sites were selected based on recent sightings of bison and physical evidence of bison (e.g., dung, wallows, hoofprints) in the area. To mimic bison foraging patterns, only terrestrial above-ground plant parts were collected. For the same reason, graminoids were prioritized for collection and sampled relative to their abundance at each location. Since only two sampling sites were wetlands and sedge cover was sparse, the sample size for sedges is low. Plants were identified to genus or species with reference to Johnson et al. (1995).

Figure 2 Selected plant sampling locations in the plains bison (P) and wood bison (W) sections of Elk Island National Park, including open (P8, W1), wet (P7, W2) and closed (P5, W8) areas.

All samples were air-dried and ground to a fine powder with a Wig-L-Bug device prior to isotopic analysis. Most of the samples (n = 131) were homogenized into ‘whole plant’ samples such as might be consumed by a relatively indiscriminate herbivore, including varying proportions of leaves, stems, seeds, and/or flowers (Table 2). To test for isotopic differences among plant parts, leaves and seeds/flowers were analyzed separately for a subset of samples (n = 34). Grass leaves are wrapped around stems before diverging as a separate blade, making stems and leaves difficult if not impossible to separate in bulk samples. Grass flowers are complex structures that include a rachis and many tiny pedicels which are likewise difficult or impossible to separate from the floret. As a result, grass leaf and seed/flower samples include variable proportions of these other tissues as well.

Table 2 Elk Island National Park plant data. An Excel version of this table is available as a Supplemental File.

Site	Habitat	Season	Sample	Taxon	Type	Parts*	δ13C	δ15N	%C	%N	C/N	
P1	Open	Summer	P1–10a	Ranunculus sp.	forb	St, L, F	−28.0	−1.0	40.4	1.3	37	
P1	Open	Summer	P1–11a	Trifolium hybridium	forb	St, L	−30.6	−1.3	37.9	2.6	17	
P1	Open	Summer	P1–12a	Fragaria virginiana	forb	St, L	−28.2	−2.0	39.6	1.8	26	
P1	Open	Summer	P1–13a	Poaceae	grass	St, L	−28.4	−1.2	38.5	1.4	32	
P1	Open	Summer	P1–14a	Poa pratensis	grass	St, Sd	−26.3		39.0			
P1	Open	Summer	P1–15a	Poaceae	grass	St, L, Sd	−26.3	−1.8	40.3	1.6	30	
P1	Open	Summer	P1–1a	Poa pratensis	grass	St, Sd	−27.7		38.8			
P1	Open	Summer	P1–1b	Poa pratensis	grass	L	−29.0	−2.6	38.9	2.1	22	
P1	Open	Summer	P1–2a	Hedysarum alpinum	forb	St, L, F	−28.3	−0.1	40.1	2.9	16	
P1	Open	Summer	P1–3a	Rubus pubescens	shrub	St, L	−27.9	−1.5	40.6	2.3	20	
P1	Open	Summer	P1–4a	Phleum pratense	grass	St, L, Sd	−28.4	−2.6	40.4	1.4	34	
P1	Open	Summer	P1–5a	Salix sp.	shrub	L	−30.0	0.0	37.7	2.0	22	
P1	Open	Summer	P1–6a	Amelanchier alnifolia	shrub	St, L, Fr	−25.9	−2.2	43.9	1.2	42	
P1	Open	Summer	P1–7a	Ranunculus sp.	forb	St, L, F	−28.0	−0.7	40.4	1.7	28	
P1	Open	Summer	P1–8a	Galium boreale	forb	St, L	−28.6	−0.7	40.3	1.6	29	
P1	Open	Summer	P1–9a	Apocynum androsaemifolium	forb	St, L, F	−29.3	+1.6	42.4	1.8	28	
P2	Closed	Summer	P2–1a	Agrimonia striata	forb	St, L	−30.4	−2.3	37.8	1.8	24	
P2	Closed	Summer	P2–2a	Salix sp.	shrub	L	−28.7	+1.6	41.1	2.0	25	
P2	Closed	Summer	P2–3a	Lonicera involucrata	shrub	L	−29.9	−1.2	38.5	1.6	28	
P2	Closed	Summer	P2–4a	Poaceae	grass	St, L	−30.3	−0.5	36.4	1.4	31	
P2	Closed	Summer	P2–5a	Poaceae	grass	St, L	−29.4	+1.5	36.9	1.3	33	
P2	Closed	Summer	P2–6a	Thalictrum sp.	forb	St, L	−30.5	+1.9	39.6	2.1	22	
P2	Closed	Summer	P2–7a	Sanicula marilandica	forb	St, F	−30.5	+0.7	38.8	2.3	20	
P2	Closed	Summer	P2–7b	Sanicula marilandica	forb	L	−32.3	−0.1	38.6	3.2	14	
P2	Closed	Summer	P2–8a	Poa pratensis	grass	St, L, Sd	−31.0	+1.1	36.6	1.6	27	
P3	Open	Summer	P3–1a	Poaceae	grass	L	−27.8	+1.6	38.2	2.1	22	
P3	Open	Summer	P3–1b	Poaceae	grass	Sd	−26.9	+1.6	39.7	2.2	21	
P3	Open	Summer	P3–2a	Poaceae	grass	St, L	−27.8	+1.3	35.9	1.8	23	
P3	Open	Summer	P3–3a	Trifolium sp.	forb	St, L, F	−28.4	−0.4	37.0	3.0	15	
P3	Open	Summer	P3–4a	Astragalus sp.	forb	St, L	−27.3	+0.7	38.1	4.8	9	
P3	Open	Summer	P3–5a	Cyperaceae	sedge	L	−27.4	+2.1	38.2	1.7	26	
P3	Open	Summer	P3–5b	Cyperaceae	sedge	Sd	−26.6	+1.7	39.2	1.9	24	
P4	Open	Summer	P4–1a	Poaceae	grass	Sd	−27.6	−1.3	39.7	2.2	21	
P4	Open	Summer	P4–1b	Poaceae	grass	L	−28.5	−3.1	36.2	2.0	21	
P4	Open	Summer	P4–2a	Bromus sp.	grass	St, Sd	−27.3	−3.2	39.2	1.4	32	
P4	Open	Summer	P4–3a	Poa pratensis	grass	St, Sd	−25.7	−0.7	38.6	1.1	42	
P4	Open	Summer	P4–3b	Poa pratensis	grass	St, L	−28.7	−3.6	37.4	1.0	43	
P4	Open	Summer	P4–4a	Poaceae	grass	St, L	−28.3	−0.4	38.5	2.2	20	
P4	Open	Summer	P4–6a	Bromus sp.	grass	Sd	−26.2	+0.1	40.5	1.8	26	
P4	Open	Summer	P4–6b	Bromus sp.	grass	L	−28.0	−1.4	39.6	1.9	24	
P4	Open	Summer	P4–7a	Trifolium hybridium	forb	St, L	−30.0	−1.5	35.7	2.5	17	
P4	Open	Summer	P4–7b	Trifolium hybridium	forb	St, F	−28.2	−0.6	38.0	1.9	23	
P5	Closed	Summer	P5–1a	Poa pratensis	grass	St, L, Sd	−29.6		37.0			
P5	Closed	Summer	P5–2a	Poaceae	grass	St, L	−30.3	−2.3	38.3	1.3	34	
P5	Closed	Summer	P5–3a	Bromus inermis	grass	Sd	−29.4	−0.1	40.2	1.9	24	
P5	Closed	Summer	P5–3b	Bromus inermis	grass	L	−30.4	−1.2	38.9	1.3	35	
P5	Closed	Summer	P5–4a	Agropyron sp.	grass	Sd	−31.2	−2.0	39.5	1.7	28	
P5	Closed	Summer	P5–4b	Agropyron sp.	grass	St, L	−32.2	−2.1	37.6	1.6	27	
P5	Closed	Summer	P5–5a	Phleum pratense	grass	Sd	−29.5	+8.4	39.4	1.9	25	
P5	Closed	Summer	P5–5b	Phleum pratense	grass	L	−31.1	+6.8	39.0	2.2	21	
P5	Closed	Summer	P5–5c	Phleum pratense	grass	St	−29.1	+6.5	38.4	0.9	51	
P5	Closed	Summer	P5–6a	Trifolium sp.	forb	St, L	−28.7	−2.0	34.5	2.5	16	
P5	Closed	Summer	P5–7a	Unidentified forb	forb	St, L	−32.1	−1.7	37.9	1.6	27	
P5	Closed	Summer	P5–8a	Poaceae	grass	L	−29.9	−1.3	39.0	2.3	20	
P5	Closed	Summer	P5–9a	Phalaris arundinacea	grass	Sd	−28.8	+0.7	39.4	2.1	21	
P5	Closed	Summer	P5–9b	Phalaris arundinacea	grass	L	−30.6	−0.8	37.6	2.6	17	
P5	Closed	Fall	P5N–1	Poaceae	grass	St, L	−29.9		37.1			
P5	Closed	Fall	P5N–2a	Poa sp.	grass	Sd	−29.2	−1.7	39.7	1.9	24	
P5	Closed	Fall	P5N–2b	Poa sp.	grass	St, L	−29.3		39.2			
P5	Closed	Fall	P5N–3b	Poaceae	grass	St, L	−29.3	+0.2	39.7	1.6	28	
P5	Closed	Fall	P5N–4a	Poa sp.	grass	Sd	−29.9	+4.2	39.7	1.1	41	
P5	Closed	Fall	P5N–4b	Poa sp.	grass	St, L	−29.7		39.9			
P5	Closed	Fall	P5N–5a	Poaceae	grass	L	−31.6		40.3			
P5	Closed	Fall	P5N–5b	Poa sp.	grass	Sd	−30.3	+6.7	38.9	0.8	60	
P6	Open	Summer	P6–1a	Agropyron sp.	grass	St, L, Sd	−28.0	+1.6	39	1.7	27	
P6	Open	Summer	P6–1b	Agropyron sp.	grass	St, L	−28.3	+1.2	37.8	3.5	12	
P6	Open	Summer	P6–2a	Bromus inermis	grass	Sd	−27.2	+1.1	39.8	1.8	26	
P6	Open	Summer	P6–2b	Bromus inermis	grass	St, L	−28.2	+0.3	38.5	1.9	23	
P6	Open	Summer	P6–3a	Trifolium sp.	forb	St, L	−29.1	−2.0	37.8	3.1	14	
P6	Open	Summer	P6–3b	Trifolium sp.	forb	St, F	−27.6	−1.3	39.1	2.3	20	
P6	Open	Summer	P6–4a	Poa pratensis	grass	St, Sd, L	−26.9		39.5			
P6	Open	Summer	P6–5a	Poaceae	grass	St, L	−28.3	+1.9	38.8	1.1	42	
P6	Open	Summer	P6–6	Poaceae	grass	L	−28.6	−2.0	38.2	4.0	11	
P6	Open	Summer	P6–7a	Poaceae	grass	L	−28.8	+2.4	38.7	2.2	20	
P6	Open	Summer	P6–8a	Ranunculus sp.	forb	St, L, F	−27.9	+0.8	37.0	1.7	26	
P7	Wet	Summer	P7–1a	Beckmannia syzigachne	grass	St, L, Sd	−29.0		37.7			
P7	Wet	Summer	P7–2a	Poa pratensis	grass	St, L, Sd	−26.8		36.7			
P7	Wet	Summer	P7–2b	Poa pratensis	grass	St, L	−30.2	+2.3	37.7	2.8	16	
P7	Wet	Summer	P7–3a	Bromus inermis	grass	St, L, Sd	−26.5	−0.1	39.8	1.6	29	
P7	Wet	Summer	P7–4a	Alopecurus aequalis	grass	St, L, Sd	−28.4	+6.5	38.4	1.6	29	
P7	Wet	Summer	P7–4b	Alopecurus aequalis	grass	L	−29.1	+6.5	38.1	2.5	18	
P7	Wet	Summer	P7–5a	Calamagrostis canadensis	grass	Sd	−28.1	+2.9	38.7	1.9	24	
P7	Wet	Summer	P7–5b	Calamagrostis canadensis	grass	L	−30.0	+2.0	35.3	1.9	22	
P7	Wet	Summer	P7–6a	Poaceae	grass	St, L	−30.2	+4.8	37.1	3.0	14	
P7	Wet	Summer	P7–7a	Agropyron trachycaulum	grass	St, L, Sd	−27.2	+2.1	38.3	2.3	19	
P8	Open	Summer	P8–1a	Poaceae	grass	St, L	−28.8	+0.9	37.0	1.8	24	
P8	Open	Summer	P8–2a	Poa pratensis	grass	St, L, Sd	−25.5	−0.6	37.7	0.8	57	
P8	Open	Summer	P8–2b	Poa pratensis	grass	L	−28.6	+1.6	37.5	2.0	22	
P8	Open	Summer	P8–3a	Poaceae	grass	St, Sd	−27.0	+0.1	38.2	0.9	50	
P8	Open	Summer	P8–3b	Poaceae	grass	L	−28.7	+2.2	37.5	2.2	20	
P8	Open	Summer	P8–3c	Poaceae	grass	L	−28.9	+1.6	37.6	2.0	22	
P8	Open	Summer	P8–3d	Agropyron trachycaulum	grass	St, Sd	−27.5	+2.7	37.8	1.5	29	
P8	Open	Fall	P8N–1a	Poa sp.	grass	Sd	−27.2	−1.9	39.2	1.1	40	
P8	Open	Fall	P8N–1b	Poaceae	grass	St, L	−27.8	+1.1	39.2	0.9	52	
P8	Open	Fall	P8N–1c	Agropyron trachycaulum	grass	Sd	−26.4	+1.0	39.1	0.9	51	
P8	Open	Fall	P8N–1d	Agropyron trachycaulum	grass	St	−26.5		41.3			
P8	Open	Fall	P8N–2a	Agropyron trachycaulum	grass	Sd	−25.6	+9.9	41.2	1.2	42	
P8	Open	Fall	P8N–2b	Agropyron trachycaulum	grass	St	−25.4		42.8			
P8	Open	Fall	P8N–2c	Poaceae	grass	L	−28.7	+7.5	38.3	0.7	65	
P8	Open	Fall	P8N–3a	Poa sp.	grass	Sd	−28.5	−0.6	38.0	0.9	48	
P8	Open	Fall	P8N–3b	Poa sp.	grass	St	−27.5		40.3			
P8	Open	Fall	P8N–3c	Poaceae	grass	L	−29.0	+1.8	35.5	0.6	66	
P8	Open	Fall	P8N–4a	Agropyron trachycaulum	grass	Sd	−27.1	+4.3	41.1	1.0	48	
P8	Open	Fall	P8N–4b	Agropyron trachycaulum	grass	St	−27.9		41.4			
P8	Open	Fall	P8N–4c	Poaceae	grass	St, L	−28.5	+3.1	39.8	1.2	39	
P8	Open	Fall	P8N–5a	Poaceae	grass	L	−28.7		40.1			
W1	Open	Summer	W1–1a	Poaceae	grass	St, L	−30.4	−3.2	37.4	1.0	43	
W1	Open	Summer	W1–2a	Poa sp.	grass	St, L, Sd	−28.3	−2.7	37.3	1.0	42	
W1	Open	Summer	W1–3a	Bromus sp.	grass	St, L, Sd	−27.2	−3.9	38.2	0.9	51	
W1	Open	Summer	W1–3b	Bromus sp.	grass	Sd	−28.3	−1.6	38.7	1.7	27	
W1	Open	Summer	W1–4a	Poa pratensis	grass	St, L	−29.9	−3.8	37.6	1.3	33	
W1	Open	Summer	W1–4b	Poa pratensis	grass	St, Sd	−28.8	−3.5	38.2	1.2	37	
W1	Open	Summer	W1–5a	Phleum pratense	grass	WS	−30.8	−1.6	38.2	0.8	54	
W2	Wet	Summer	W2–1a	Poaceae	grass	St, L	−30.1	+0.4	38.2	1.8	25	
W2	Wet	Summer	W2–2a	Poa sp.	grass	St, L	−30.5	−1.5	36.6	1.6	26	
W3	Closed	Summer	W3–1a	Poaceae	grass	St, L	−31.1	−1.4	35.2	1.3	31	
W3	Closed	Summer	W3–2a	Poaceae	grass	St, L	−32.6	−2.4	36.2	0.9	47	
W3	Closed	Fall	W3N–1a	Poaceae	grass	St, L	−29.5		39.8			
W3	Closed	Fall	W3N–2a	Poaceae	grass	L	−29.3		39.6			
W3	Closed	Fall	W3N–3a	Poaceae	grass	L	−30.2	+0.7	41.0	0.8	61	
W4	Open	Summer	W4–1a	Carex atheroides	sedge	St, L, Sd	−27.9	+0.4	37.9	1.2	38	
W4	Open	Summer	W4–2a	Carex sp.	sedge	L	−29.8	+2.9	37.1	1.4	31	
W4	Open	Summer	W4–3a	Poaceae	grass	St, L	−29.7	−1.2	38.8	1.6	28	
W4	Open	Summer	W4–3b	Poaceae	grass	St, L, Sd	−29.0	−1.1	37.8	1.3	35	
W4	Open	Summer	W4–4a	Carex atherodes	sedge	St, L, Sd	−27.7	+0.9	38.7	1.3	35	
W4	Open	Summer	W4–4b	Carex atherodes	sedge	St, L	−27.9	−0.5	37.6	1.1	39	
W5	Open	Summer	W5–1a	Poa sp.	grass	St, Sd	−26.9	−1.0	38.4	1.0	46	
W5	Open	Summer	W5–1b	Poa sp.	grass	L	−27.9	+0.5	38.6	2.2	20	
W5	Open	Summer	W5–2a	Agropyron sp.	grass	St, L, Sd	−26.4	+1.7	38.7	2.1	22	
W5	Open	Summer	W5–2b	Agropyron sp.	grass	L	−28.8	+1.2	37.7	2.4	18	
W5	Open	Summer	W5–3a	Poaceae	grass	St, L	−28.2	+2.5	38.2	3.6	12	
W6	Open	Summer	W6–1a	Phleum pratense	grass	St, L, Sd	−29.2	−0.5	37.9	1.4	32	
W6	Open	Summer	W6–1b	Phleum pratense	grass	L	−26.6	+2.9	38.5	2.8	16	
W6	Open	Summer	W6–2a	Poa sp.	grass	St, L, Sd	−29.7	−2.2	38.6	0.9	50	
W6	Open	Summer	W6–2b	Poa sp.	grass	St, L	−30.4	−0.3	36.3	1.7	25	
W7	Open	Summer	W7–1a	Equisetum sp.	horsetail	St, L	−29.6	+4.5	30.7	1.6	22	
W7	Open	Summer	W7–2a	Poa sp.	grass	Sd	−27.7	−0.9	37.4	1.2	36	
W7	Open	Summer	W7–2b	Poa sp.	grass	St, L	−28.6	−0.1	37.2	1.4	31	
W7	Open	Summer	W7–3a	Phleum pratense	grass	Sd	−26.7	−1.2	39.5	1.5	31	
W7	Open	Summer	W7–3b	Phleum pratense	grass	St, L	−28.1	−2.4	38.1	1.1	40	
W8	Closed	Summer	W8–1a	Agropyron sp.	grass	St, L	−29.7	−1.9	37.9	1.5	29	
W8	Closed	Summer	W8–2a	Bromus sp.	grass	St, Sd	−28.4	+6.5	39.7	2.1	22	
W8	Closed	Summer	W8–3a	Equisetum sp.	horsetail	WS	−30.6	+3.2	32.7	1.7	23	
W8	Closed	Summer	W8–4a	Poaceae	grass	L	−29.6	−3.9	37.6	0.9	47	
W8	Closed	Summer	W8–5a	Poaceae	grass	St, L	−29.8	−0.6	38.5	2.0	23	
W8	Closed	Summer	W8–6a	Poaceae	grass	L	−28.3	−1.0	36.9	1.1	38	
W8	Closed	Summer	W8–7a	Bromus sp.	grass	Sd	−29.1	−0.6	40.4	2.1	23	
W8	Closed	Summer	W8–7b	Bromus sp.	grass	St, L, Sd	−28.8	−1.4	39.6	1.6	29	
W8	Closed	Summer	W8–8a	Poaceae	grass	St, L, Sd	−28.2		38.1			
W9	Open	Summer	W9–1a	Poa sp.	grass	St, Sd	−26.9		38.2			
W9	Open	Summer	W9–2a	Poaceae	grass	L	−28.1	−0.9	38.2	1.4	33	
W9	Open	Summer	W9–3a	Melilotus officinalis	forb	St, L, F	−27.7	−1.2	37.6	2.4	18	
W9	Open	Summer	W9–4a	Equisetum sp.	horsetail	WS	−29.1	+5.1	33.0	2.0	19	
W9	Open	Summer	W9–5a	Poaceae	grass	St, L	−28.4	+0.7	38.0	1.5	31	
W9	Open	Summer	W9–6a	Bromus sp.	grass	St, L, Sd	−25.4		39.0			
W9	Open	Summer	W9–7a	Poaceae	grass	St, L, Sd	−26.9	−0.3	37.9	1.4	31	
W9	Open	Summer	W9–8a	Trifolium hybridium	forb	St, F	−26.3	−1.2	38.0	2.0	22	
W9	Open	Summer	W9–8b	Trifolium hybridium	forb	St, L	−27.9	−1.4	36.4	3.4	13	
W9	Open	Fall	W9N–1a	Poa sp.	grass	Sd	−26.7	−0.9	40.1	0.9	51	
W9	Open	Fall	W9N–1b	Poa sp.	grass	St, L	−26.2		40.6			
W9	Open	Fall	W9N–2a	Poaceae	grass	St, L	−28.1		38.9			
W9	Open	Fall	W9N–3a	Poaceae	grass	Sd	−27.1	+5.7	41.9	1.0	49	
W9	Open	Fall	W9N–3b	Poaceae	grass	St	−24.9		43.8			
W9	Open	Fall	W9N–4a	Bromus sp.	grass	Sd	−25.6	+2.2	40.4	1.4	35	
W9	Open	Fall	W9N–4b	Bromus sp.	grass	St, L	−25.3		41.9			
Notes:

* St, stem; L, leaf; F, flower; Fr, fruit; Sd, seed; WS, whole sample.

Many grass samples lacked an inflorescence, making more specific identification difficult or impossible.

Carbon and nitrogen isotope measurements

Carbon and nitrogen isotope values (δ13C, δ15N) and carbon and nitrogen contents (dry weight %C, %N) were obtained using an Elementar VarioMicro Cube elemental analyzer coupled with an Isoprime isotope-ratio mass spectrometer in continuous-flow mode. Carbon and nitrogen isotope values were obtained during the same run by combusting approximately 1 mg of sample and using a high level of dilution to reduce the carbon dioxide gas peaks. Nitrogen isotope results from samples with N2 peaks <1 nA were excluded unless duplicate analyses exhibited similar reproducibility to samples with larger gas peaks. Carbon isotope values of the low-nitrogen samples were retained since the carbon peaks were large enough to produce reliable results. The samples with low nitrogen-gas peaks are those lacking δ15N values in Table 2.

δ13C values were calibrated to VPDB and δ15N values were calibrated to AIR using USGS-40 and USGS-41 or 41a (accepted δ13C values of −26.39, +37.63 and +36.55‰ and accepted δ15N values of −4.52, +47.57 and +47.55‰, respectively). Sample replicates (minimum 10% of samples in each run) and internal check standards of methionine, amaranth, and red lentil (long-term mean δ13C of −28.60, −13.59, −26.12‰; long-term mean δ15N of −5.04, +2.94 and −1.09‰, respectively) were used to monitor measurement uncertainty. Uncertainty measures were calculated following the method of Szpak, Metcalfe & Macdonald (2017). For δ13C, precision u(Rw) was 0.11‰, accuracy (u(bias)) was 0.09‰, and total analytical uncertainty (uc) was 0.14‰. For δ15N, precision was 0.23‰, accuracy was 0.23‰, and total analytical uncertainty was 0.33‰.

Statistical analyses

Statistical analyses were conducted using Excel for Office 365 and PAST (PAleontological STatistics) 4.03. Shapiro–Wilk W tests were used to assess the normality of distributions. Levene tests were used to evaluate the homogeneity of variance. Normally distributed datasets (carbon isotope values) were compared using Student’s t-tests (two independent samples), paired-sample t-tests (two paired samples), or one-way ANOVA F-tests with Tukey’s post-hoc comparisons (three or more independent samples). Non-normally distributed datasets (nitrogen isotope values) were compared using Mann–Whitney U tests (two independent samples), Wilcoxon sign-rank tests (two paired samples) or Kruskal–Wallis H tests with Dunn-Bonferroni post-hoc comparisons (three or more independent samples). Alpha was set to 0.05 for all statistical comparisons. In the text below, means are reported with standard deviations, unless noted otherwise.

Results

Whole sample

Plant δ13C values ranged from −32.6 to −24.9‰, with a mean and standard deviation of −28.5 ± 1.5‰ (Tables 2, 3). Plant δ15N values ranged from −3.9 to +9.9‰, with a mean and standard deviation of +0.4 ± 2.7‰. The shape of the distribution was normal for δ13C (Shapiro−Wilk W = 0.99, n = 165, p = 0.7; skewness = −0.05) and positively skewed for δ15N (Shapiro–Wilk W = 0.92, n = 141, p < 0.001; skewness = 1.14) (Fig. 3).

Figure 3 EINP plant carbon and nitrogen isotope distributions.

Table 3 Summary statistics for EINP plant samples grouped all together (whole sample) and by type, habitat, parts, and season of collection.

	δ13C (‰, VPDB)	δ15N (‰, AIR)	
	n	Mean	Median	SD	Range	n	Mean	Median	SD	Range	
 Whole Sample	165	−28.5	−28.5	1.5	−32.6 to −24.9 (7.7)	141	+0.4	−0.3	2.7	−3.9 to +9.9 (13.8)	
Type	
 Grass	128	−29.0	−28.5	1.5	−32.6 to −24.9 (7.7)	104	+0.5	−0.1	2.9	−3.9 to +9.9 (13.8)	
 Sedge	6	−27.9	−27.8	1.1	−29.8 to −26.6 (3.2)	6	+1.3	+1.3	1.2	−0.5 to +2.9 (3.4)	
 Shrub	5	−28.5	−28.7	1.7	−30.0 to −25.9 (4.1)	5	−0.7	−1.2	1.5	−2.2 to +1.6 (3.8)	
 Forb	23	−29.0	−28.4	1.5	−32.3 to −26.3 (6.0)	23	−0.7	−1.0	1.2	−2.3 to +1.9 (4.2)	
 Horsetail	3	−29.8	−29.6	0.8	−30.6 to −29.1 (1.5)	3	+4.3	+4.5	0.6	+3.2 to +5.1 (1.9)	
Habitat	
 Open	108	−27.9	−28.0	1.2	−30.8 to −24.9 (5.9)	94	+0.1	−0.4	2.4	−3.9 to +9.9 (13.8)	
 Closed	45	−30.0	−29.9	1.1	−32.6 to −28.2 (4.4)	37	+0.5	−0.6	3.1	−3.9 to +8.4 (12.3)	
 Wet	12	−28.8	−29.1	1.4	−30.5 to −26.5 (4.0)	10	+2.6	+2.2	2.7	−1.5 to +6.5 (8.0)	
Parts (same plant)	
 Leaf	34	−29.0	−28.8	1.5	−32.3 to −25.3 (7.0)	28	+0.5	−0.1	2.8	−3.6 to +7.5 (11.1)	
 Seed	34	−27.8	−27.7	1.5	−31.2 to −25.5 (5.7)	28	+1.0	0.0	3.0	−2.2 to +9.9 (12.1)	
Season	
 Summer (all)	133	−28.7	−28.5	1.5	−32.6 to −25.4 (7.2)	124	+0.1	−0.5	2.4	−3.9 to +8.4 (12.3)	
 Summer (match*)	32	−29.0	−28.9	1.9	−32.6 to −25.4 (7.2)	29	+0.5	−0.6	2.9	−2.4 to +8.4 (10.0)	
 Fall	32	−28.0	−28.3	1.7	−31.6 to −24.9 (6.7)	17	+2.5	+1.8	3.4	−1.9 to +9.9 (11.8)	
Note:

* Summer (match) excludes data from locations that were not sampled in Fall.

Plant types

The mean δ13C values of grasses, sedges, shrubs, forbs and horsetail were within 1.9‰ of one another (Table 3), and an ANOVA showed no statistically significant differences among the groups (F(4,160) = 1.3, p = 0.28). With horsetail removed (because of its small sample size), there were still no significant differences in δ13C among grasses, sedges, shrubs, and forbs (F(3,158) = 1.0, p = 0.39) There was a significant difference among the δ15N values of plant types (H(4) = 12.9, p = 0.01), but the Dunn–Bonferroni test suggested that only the horsetail-forb comparison was significant (p = 0.03). With horsetails removed there was no statistically significant difference among grasses, sedges, shrubs, and forbs (H(3) = 7.0, p = 0.07), and their medians were within 2.3‰ of one another. Although the median grass δ15N value did not significantly differ from that of any other group, grasses had the greatest variability of any plant type, and grass samples had both the highest (>+5.1‰) and lowest (<−2.3‰) individual plant δ15N values (Table 3, Fig. 4). A Levene’s test from medians (i.e., Brown–Forsythe test) indicated that the difference in the variability of δ15N among plant types was statistically significant (p = 0.01).

Figure 4 Carbon and nitrogen isotope compositions of EINP plants grouped by type.

Habitats

Plant growth habitat had a significant effect on the carbon isotope compositions of plants (F(2,162) = 48.8, p < 0.001). The differences among all three groups were statistically significant, with the highest δ13C values in open areas (−27.9 ± 1.2‰, n = 108), intermediate values in wet areas (−28.9 ± 1.4‰, n = 12) and the lowest values in closed-canopy areas (−30.0 ± 1.1‰, n = 45) (Table 3, Fig. 5). Growth habitat also affected δ15N values (H(2) = 7.7, p = 0.02), with higher δ15N values in wet habitats (+2.6 ± 2.7‰, n = 10) compared to those in either open areas (+0.1 ± 2.4‰, n = 94) or closed canopy areas (+0.5 ± 3.1‰, n = 37). Although wet areas had higher mean (and median) δ15N values than the open or closed-canopy areas, the latter two habitat types hosted the plants with the highest individual δ15N measurements (Fig. 5). As mentioned previously, these extreme δ15N values were all from grass samples. There was a positive skew in the δ15N values of plants from open environments (W = 0.9, n = 94, p < 0.001) and closed environments (W = 0.9, n = 37, p < 0.001).

Figure 5 Plant carbon and nitrogen isotope distributions by growth habitat.

The box encloses the interquartile range and median (horizontal line). The whiskers represent the full range of measured values.

Plant parts

Carbon isotope compositions of leaves were on average 1.2‰ lower than those of seeds/flowers from the same plants (paired samples t = 7.8, df = 33, p < 0.001) (Table 3). Furthermore, the great majority of individual plant samples had lower leaf than seed/flower δ13C values, with seed/flower minus leaf differences (Δ13Cseed–leaf) of individual plants ranging from −0.5 to +3.1‰ (Fig. 6). The lowest mean and individual δ13C values were obtained from leaves in closed habitats, and the highest mean δ13C from seeds in open habitats (Fig. 6).

Figure 6 Differences between the carbon and nitrogen isotopic compositions of seeds and leaves from the same plants.

Nitrogen isotope compositions of leaves were 0.5‰ lower on average than those of seeds/flowers from the same plants (Table 3), but the difference was not statistically significant (Wilcoxon W = 250, df = 27, p = 0.06). Individual plants had highly variable seed-minus-leaf differences (Δ15Nseed–leaf), ranging from −2.4 to +2.9‰ (Fig. 7).

Figure 7 Comparison of carbon and nitrogen isotope compositions and nitrogen contents of EINP plants collected from matched locations in summer (late June) and fall (early November).

Seasonal changes

Seasonal shifts in plant δ13C and δ15N occurred between early summer (late June) and mid fall (early November) (Table 3, Fig. 7). Plant δ13C increased slightly during fall, both for the whole dataset (t(163) = 2.1, p = 0.04, mean difference of 0.6‰) and when only locations sampled in both seasons were included (t(62) = 2.2, p = 0.03; mean difference of 1.0‰). Plant δ15N also increased during fall, both for the whole dataset (U = 582.5, p = 0.003; mean difference of 2.5‰) and when only samples from matched locations were compared (U = 145, p = 0.02; mean difference of 2.0‰). Plant nitrogen contents (%N) also significantly decreased from summer to fall (whole sample: U = 265, p < 0.001; mean difference of 0.8% matched locations: U = 71, p < 0.001; mean difference of 0.7%) (Fig. 7). The true seasonal decrease in plant nitrogen content is likely greater than this value implies, since proportionally more fall plant samples were excluded due to their small gas peaks (see Table 3).

Discussion

Plant isotopic distributions

The distribution of plant δ13C values was normal. The EINP whole-sample mean δ13C of −28.5‰ is somewhat lower than the modern global mean C3 plant δ13C value of −27.0‰ determined by Kohn (2010). This can be attributed to two main factors: (1) the δ13C of atmospheric CO2 during our sample collection (in 2016) was significantly lower than Kohn’s (2010) normalized value of −8.0‰ because of the ongoing effects of fossil fuel burning (Long et al., 2005), and (2) Kohn’s (2010) study excluded understory plants with δ13C values below −31.5‰, whereas the present study did not.

Distributions of plant nitrogen isotope compositions were positively skewed. Skewness of isotopic distributions is seldom explicitly evaluated, and isotopic data presentations that facilitate visual examination of skewness (e.g., frequency histograms, box-and-whisker plots) are relatively rare, so it is difficult to determine how common skewed plant nitrogen isotope distributions may be. Metcalfe & Mead (2019) observed a negatively skewed δ15N distribution for Pleistocene plants. Funck et al. (2020: Supplemental Material) provide box-plots that appear to illustrate positively skewed modern grass δ15N and negatively skewed modern herb δ15N distributions, but they did not explicitly evaluate skewness. The other plant isotopic studies reviewed here neither evaluated skewness nor presented data in forms that make it easy for readers to evaluate. Determining the shape of a distribution is often overlooked but testing for normality is a critical first step before utilizing parametric statistical methods, at least when sample sizes are small (which is typical in most archaeological and paleontological studies) (Ghasemi & Zahediasl, 2012). Failing to recognize skewed isotopic distributions can result in inappropriate data reporting (e.g., use of means and standard deviations rather than medians and interquartile ranges) and use of statistical tests whose assumptions are not met (i.e., parametric statistical tests), potentially producing invalid results and leading to erroneous interpretations. Assessing the skewness of dietary components (and other characteristics of data distribution) is also critical for studies using stable isotope mixing models, which typically assume normal distributions and require dietary inputs of means and standard deviations (Cheung & Szpak, 2020).

It is possible that skewed plant δ15N distributions could help explain the strong nitrogen isotopic niche partitioning that has been observed among herbivores in some ecosystems. In particular, mammoths tend to have significantly higher δ15N values than co-existing herbivores, which is related to a dietary (rather than physiological) difference (Schwartz-Narbonne et al., 2015). In the present study, grasses had the greatest variability in δ15N of any plant taxon and all of the most positive δ15N values in the skewed tail of the distribution (i.e., values >5.1‰) were from grasses (Table 3, Fig. 4). Grasses are the predominant food of mammoths, but also of bison, who do not have enriched δ15N values. If variables could be identified that predict which grass specimens within a given ecosystem have high δ15N values (i.e., taxa, parts, growth-stages, growth habitats), then it might be possible to determine if mammoths were likely to have been selecting such grasses (for example, by employing different feeding strategies or preferring different microhabitats). In general, a herbivore preferentially selecting plants from the skewed ‘tail’ of an isotopic distribution would be predicted to occupy a distinct isotopic niche relative to herbivores that are randomly selecting plants from throughout the distribution. This would also be true of herbivores selecting plants whose δ-values fall within the tails of a normal distribution, but a skewed plant distribution would be predicted to result in greater herbivore isotopic niche differentiation due to the more extreme values of outliers in the skewed tail of the distribution.

Plant types

The overlapping δ13C and δ15N values of grasses, sedges, forbs, and shrubs in EINP highlights the importance of understanding local plant variability when interpreting herbivore isotopic compositions. Previous research has established some generalities about isotopic differences among primary producers. For example, lichens often have higher δ13C values than terrestrial plants (e.g., Brooks et al., 1997; Teeri, 1981), woody gymnosperms generally have higher δ13C values than woody angiosperms (Hare & Lavergne, 2021), and aquatic plants tend to have higher δ15N values than terrestrial plants (e.g., Kielland, 2001; Plint, Longstaffe & Zazula, 2019. However, comparisons of differences among plant types at local levels can produce disparate results (e.g., Drucker et al., 2010: Fig. 4), which is perhaps not surprising when one considers the complex range of environmental factors that affect δ13C and δ15N, as well as the fact that researchers select different plant groups for study and even categorize them differently (Table 1). Compilations of isotopic data from plants growing in various habitats (i.e., global or regional datasets) can obscure the effects of microhabitats (e.g., degree of canopy cover, altitude, aridity, etc.), which may be more important variables for interpreting herbivore isotopic compositions than plant type. Studies that compare herbivore isotopic compositions in ancient C3 ecosystems to a plant baseline organized by plant type (e.g., Schwartz-Narbonne et al., 2019; Schwartz-Narbonne et al., 2021) presuppose that type is the most important predictor of a plant’s isotopic compositions. An alternative approach is to put equal or greater emphasis on major environmental factors that influence plant isotopic compositions, such as the canopy effect (e.g., Drucker et al., 2008; Hofman-Kamińska et al., 2018) or ecosystem changes (e.g., Drucker et al., 2011; Metcalfe & Longstaffe, 2014).

The plant-type data in the present study highlight the importance of ecosystem-specific contexts. In particular, it is not appropriate to assume that grasses, sedges, shrubs and forbs will have consistent relative isotopic differences in disparate environments and temporal intervals. Consequently, similar isotopic niches among animals does not necessarily mean that animals ate the same things, or that “one species could fulfill another’s ecological role” (Schwartz-Narbonne et al., 2019: 1). Rather, isotopic niche overlap could simply indicate that there are minimal isotopic differences among the disparate plants consumed by herbivores in that environment. Likewise, minimal isotopic variations in serially-sampled animal tissues does not necessarily suggest that animals had highly specialized diets with minimal variation. On the contrary, minimal seasonal isotopic variations in herbivore tissues could occur even when animals undertake significant seasonal changes in diet if there are no significant differences among plant types in that area. Given these complexities, the key to being able to make meaningful interpretations of herbivore isotopic compositions is to have a good understanding of which isotopic baselines and variables are most important for a particular study and to seriously consider alternative interpretations based on the various factors that can influence isotopic systems. In general, isotopic niches are far from equivalent to dietary niches or dietary specializations.

Plant parts and habitats: carbon isotopes

Lower plant δ13C values in EINP closed habitats compared to open habitats (~2‰ on average) is consistent with the well-known canopy effect, in which understory plants have significantly lower δ13C values than plants that make up the canopy or emergent layers, or plants that grow in open areas (e.g., Bonafini et al., 2013; Chevillat et al., 2005; Drucker et al., 2008; Van Der Merwe & Medina, 1991). The lower δ13C values in EINP leaves relative to seeds/flowers (~1‰ on average) is likewise in agreement with the 1–3‰ difference that has been reported in many other studies (e.g., Badeck et al., 2005; Ghashghaie & Badeck, 2014; Metcalfe & Mead, 2019).

The EINP plant isotopic data suggest that among herbivores, a combined effect of plant-part and habitat-selection could result in significant carbon isotope niche partitioning within C3 environments, with the largest differences between animals consuming seedy/flowery plants in open environments (higher δ13C) and those selecting seedless/flowerless plants in closed environments (lower δ13C). This offers an alternative to assuming that animal niche partitioning in C3 environments is due to differing proportions of grass vs browse or consumption of different plant taxa. Many previous studies have used herbivore δ13C to infer the ‘openness’ of utilized habitats (e.g., Bocherens et al., 2015; Doppler et al., 2017; Drucker et al., 2003; Drucker et al., 2011), but few have considered the additional isotopic effects of plant-part differences, such as the decrease in leaf δ13C than occurs as the leaf expands (Vogado et al., 2020) or differences among seedier versus seedless plant parts (but see Guiry et al., 2020 for an exception). The effects of ‘seedy’ vegetation on herbivore isotopic compositions deserves further study, since there may also be differential digestibility among seeds and leaves that influences their incorporation into herbivore tissues.

In general, herbivore feeding specializations go beyond selection of particular plant forms, species and habitats to include specialization on particular plant parts and growth stages. These differential feeding strategies might have particularly pronounced isotopic effects in an environment like the mammoth steppe, where co-existing grazers likely consumed different parts of the same plants. For example, elephantids rip out tall (potentially seedy) bunches of grasses by grabbing them with their trunks, whereas bison break off short (probably less seedy) grasses and tall/mid-level new growth with their tongues and teeth (Guthrie, 1982). On the mammoth steppe, bison tended to have higher δ13C values than mammoths in a range of locations and temporal intervals (e.g., Bocherens, 2015). Higher δ13C values in a taxon that consumes shorter grasses is the opposite of what would be expected if ‘seediness’ was a factor in isotopic niche differentiation. However, the higher δ13C values of bison could result from bison consuming a larger proportion of short, newly-grown leaves, which tend to have higher δ13C values than older mature leaves (Vogado et al., 2020). Regardless of what drives isotopic niche differentiation on the mammoth steppe, the results of the present study suggest that in some environments, habitat and plant-part selection could have greater isotopic effects on herbivore isotopic compositions than selection of different plant taxa.

Plant habitat: nitrogen isotopes

EINP plants from the wet habitat tended to have higher δ15N values than plants from the dry (open or closed-canopy) environments. Although this contrasts with the general trend towards higher δ15N values in drier locations that is often observed on regional and global scales (Craine et al., 2009; Handley et al., 1999; Wang et al., 2014), it is consistent with the higher plant δ15N values often observed in aquatic systems relative to terrestrial systems (e.g., Cloern, Canuel & Harris, 2002; Kielland, 2001; Plint, Longstaffe & Zazula, 2019. It is possible (and perhaps likely) that EINP terrestrial plants growing in seasonally wet areas obtained some nitrogen from aquatic sources, leading to higher δ15N values. It is also possible that herbivore dung is frequently deposited in wetland areas when animals come to drink, contributing 15N-enriched nitrogen to the wetland system and mimicking the established effects of manuring on plant δ15N (e.g., Bogaard et al., 2007; Szpak et al., 2014). It is important to note that the sample size available for EINP wetland habitats was small, so the reliability of this habitat difference should be re-examined in future studies. Nevertheless, in combination with previous studies that clearly show higher δ15N values among aquatic plants, these results suggest caution for archaeologists and paleoecologists who interpret higher herbivore δ15N as indicators of increased aridity. An alternative explanation (among others) for high herbivore δ15N values could be the consumption of plants growing in or near nutrient-rich wetlands.

Seasonal changes in plant isotopic compositions

A summer-to-fall (late June to early November) increase in both δ13C and δ15N (by ~1 and 2‰, respectively) was observed in EINP plants. This could be due to a combination of factors, including changes in the biochemical compositions of tissues, changes in source C and N isotopic compositions, remobilization of nutrients into roots for winter, and early decomposition. The direction and magnitude of seasonal isotopic changes in plants may vary among environments and locations. For example, Karlsson, Eckstein & Weih (2000) found that the δ15N values of most Subarctic plants in northern Sweden increased between the snowmelt (May) and mid-June, but decreased in August and September, with a range in seasonal variation of 2.1 to 5.3‰. On the other hand, the timing of key seasonal changes (e.g., temperature increases and decreases) varies considerably among locations and makes seasonal generalizations challenging.

Reconstructing ecosystem-specific seasonal changes in plant δ13C and δ15N could help researchers interpret serial-sampling studies of herbivore isotopic compositions, which may vary due to seasonal changes in diet, physiology, and/or isotopic variations in plants. Seasonal changes in the diets of a range of herbivores have been studied within C3-dominated ecosystems, and these changes are often relatively small in magnitude (~2 to 3‰ or less). For example, Funck et al. (2020) observed temporal changes in sectioned wood bison (Bison bison athabascae) hair δ13C and δ15N that they attributed to nutritional stress. Julien et al. (2012) serially-sampled steppe bison (Bison priscus) teeth and interpreted small winter increases in δ13C as an indication of lichen consumption. Metcalfe & Longstaffe (2014) identified different seasonal patterns in the tooth enamel of mastodons (Mammut americanum) that lived in the same geographical area during different time periods, which they suggested were the result of major vegetational shifts. Kielland (2001) serially-sampled Alaskan moose (Alces alces) hooves and interpreted variations of about 2–3‰ as evidence for seasonal changes in diet. Plant isotopic values and their variability underlie the interpretations of all these studies.

Bison generally consume graminoids year-round but may seasonally switch between grasses and sedges, and/or consume forbs and woody plants when graminoids are not available (Gogan et al., 2010). The minimal isotopic differences among plant taxa in EINP suggests that these seasonal shifts in bison foraging strategies might not be recorded in the isotopic compositions of incrementally growing bison tissues such as teeth or hair. However, based on the EINP seasonal plant data, one might predict that seasonal isotopic shifts in the plants themselves could be recorded in serially-sampled bison tissues. Generalizing to other environments, researchers should be aware that seasonal changes in herbivore isotopic compositions do not necessarily indicate changes in foraging strategies, but can result from isotopic changes within the plants themselves.

Conclusions

This study has provided a plant carbon and nitrogen isotope baseline for future studies of herbivores in Elk Island National Park, and for archaeological and paleontological studies of animals in C3-dominated environments. A strong positive skew to the plant nitrogen isotope distributions highlights the need for isotopic researchers to explicitly evaluate the characteristics of their distributions (e.g., normal versus skewed) so that they can select appropriate measures of central tendency and variability, conduct appropriate statistical tests, and/or utilize isotopic mixing models.

In this study no statistically significant differences were observed in the δ13C or δ15N of the majority of C3 plant types (grasses, sedges, forbs, and shrubs), but there were differences among plant parts, habitats, and seasons. These results carry three important implications. First, animals consuming different plant taxa could have similar or identical isotopic compositions. Second, animals consuming the same C3 plant taxa could have different isotopic compositions if they select plants growing in different habitats (e.g., open, closed, wet) and/or different plant parts (e.g., leaves, seeds). Third, seasonal changes in herbivore isotopic compositions need not indicate a shift in foraging strategy, but rather could result from seasonal isotopic changes within dietary plants. Based on first principles of isotope systematics, these conclusions are not new. However, too often isotopic niche partitioning is equated with dietary niche partitioning, and a lack of isotopic niche partitioning is taken to reflect similar or identical diets. It is critical that researchers bear in mind the complexities of isotopic systems when making paleodietary inferences, and support their interpretations with explicit independent lines of evidence on plants and animals (i.e., isotopic baselines) in relevant ecosystems and at appropriate scales of analysis.

Supplemental Information

Supplemental Information 1 Elk Island National Park plant data.

Click here for additional data file.

Thank you to Vandy Bowyer for assistance in sample collection and GPS recording, Wes Olson and Johane Janelle for fieldwork assistance and valuable discussions, Jack Brink, Pinette Robinson, and Dustin Guedo for assistance with permits and background information, Slade and Henry Loutet for assistance with sample organization, Evangeline Bell and Georgia de Rappard-Yuswack for laboratory assistance, Peter Demontigny and Scott Hamilton for mapping assistance, Michael Blake and the UBC Department of Anthropology for laboratory access, and Jack Ives for making introductions critical to the success of this project.

Additional Information and Declarations

Competing Interests

Author Contributions

Field Study Permissions

Data Availability

The author declares that she has no competing interests.

Jessica Z. Metcalfe conceived and designed the experiments, performed the experiments, analyzed the data, prepared figures and/or tables, authored or reviewed drafts of the paper, and approved the final draft.

The following information was supplied relating to field study approvals (i.e., approving body and any reference numbers):

Sample collection was approved by Parks Canada (Research and Collection Permit EI-2016-21863).

The following information was supplied regarding data availability:

The full dataset is available in Table 2 and the Supplemental File.

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
