# Peer review of "C3 plant isotopic variability in a boreal mixed woodland: implications for bison and other herbivores"

_PeerJ, doi:10.7717/peerj.12167_

## Round 0.1 · original submission · Minor Revisions

Firstly, thanks for your patience. I've been in the field without internet access for the last while and so this could have been back with you a bit earlier ​— but it is what it is.

Secondly, all reviewers were very positive about the important nature of your study. They provide a range of valuable suggestions, which I believe will improve your already well-written manuscript. Thankfully, none of the reviewers have picked up a dreaded "fatal flaw" (that requires deep contemplation or a return to the drawing board)​— and so I look forward to seeing your revised manuscript in the near future.

Regards,
Alastair

·

Basic reporting

No comment.

Experimental design

Good reporting of experimental methods.

Validity of the findings

No Coment

Additional comments

This work presents an isotopic survey of the ecology of a C3 ecosystem, and a discussion of the relevance of the findings for herbivore ecology. I especially like the finding “seasonal changes in ... isotopic compositions do not necessarily indicate changes in foraging strategies, but can result from isotopic changes within the plants themselves.” This is indeed an under-appreciated aspect of many herbivore isotope studies, so I welcome a more integrated and cross-disciplinary approach.

I have a few comments and suggestions, which I hope you will find useful.

Line 22-23. “These results suggest that (1) herbivores eating these different types of C3 plant can have identical tissue isotopic compositions”. Whilst the Abstract is generally clear, this concluding sentence is a little confusing to me, because it comes immediately after the sentence (line 21-22) which states that there is seasonal variability in δ13C and δ15N.
If I have fully understood the intended meaning here, perhaps a more precise phrase would be: “(1) in this ecosystem, dietary preference for C3 plant type should have little effect on tissue isotopic compositions of herbivores, within a particular season and habitat” or something similar? Then, go on to the effects of seasonality and habitat (point 2). Unless you are directly testing this against herbivore δ13C, it might be better to make it clearer that these are predictions – not findings?

As it is currently written, the possible implications for herbivores are a major focal point of the paper, but the paper is a bit short of studies of intra-tooth herbivore and bison δ13C (i.e. data). Therefore, if the conclusions and suggestions are to remain, I think that the work would benefit from a greater incorporation of previously-published Bison intra-tooth δ13C data into the discussion (i.e. how do the findings transfer into herbivore C-isotope space, with tissue-diet enrichment, and explain certain trends in previous data?). This would reinforce the present emphasis on implications for higher trophic levels. Some very nice references are briefly mentioned in the Introduction (lines 41-47), but they are not mentioned again. Fleshing these refs - and data therein - out a bit in later discussion subsections will help to tie the author’s predictions to the paper.

It is interesting that leaves are more negative for δ13C than seeds/flowers. The author might find the recent paper of Vogado et al (2020) of relevant to this finding, as it presents a physiological reason for this effect (e.g. incorporation of older C-pool during leaf aging), and these authors found a difference of around 2 per mil. Vogado, N.O., Winter, K., Ubierna, N., Farquhar, G.D. and Cernusak, L.A., 2020. Directional change in leaf dry matter δ 13C during leaf development is widespread in C3 plants. Annals of Botany, 126(6), pp.981-990. This reference could be added to that of Lowdon & Dyck (1974), in Line 59.

Lines 253-254 “Previous studies have also found no isotopic differences, or variable differences, among C3 plant life-forms in different environments (e.g., Barnett 1994; Chen et al. 2005; Funck et al. 2020; Tahmasebi et al. 2017).” Bear in mind that this might be true amongst particular C3 plant groups for terrestrial systems, but is not true between C3 aquatic/marine systems, or between major macro-evolutionary C3 lineages (in different environmental conditions). See Porter et al (2018) GCA, Leavitt & Newberry (1992) Plant Physiol., also Hare & Lavergne (2021) GCA, and Pedentchouk et al. (2008) Organic Geochem.

Lines 77-78 “This is especially true in mid- and high-latitude woodland environments, where botanical remains tend to degrade rapidly due to high soil acidity.” As far as I understand it, the opposite is true; that preservation is generally better in high-latitude contexts, because the effect of (lower) temperature predominates diagenesis and respiration (in soils). Preservation is worst in mid-latitude contexts, because of very high soil acidity and high temperatures, and higher soil respiration. Perhaps a reference could help clarify things here?

Line 101 “All grasses in the park utilize C3 photosynthesis”. Unless there is physical evidence to support this, I think I would omit this sentence. Even in C3 temperate ecosystems it is extremely rare to have absolutely no C4 photosynthesis (e.g. some seasonal, highly localised pockets of grasses will almost certainly be C4), especially if summers are moderate. This will have little effect on herbivore δ13C.

Lines 276-277. I’m wondering here what proportion seeds will contribute to the overall herbivore isotopic composition, given that seeds are generally much tougher, and much less digestible than leaves? In fact, many plants use animals for seed dispersal – implying that they are generally undigested. Perhaps the “poop literature” might have some insights?

Table 1. For future work, another useful resource for cold-temperate plant isotope data could be found by looking through the compilation of Cornwell W. K., Wright I., Turner J., Maire V., Barbour M., Cernusak L., Dawson T., Ellsworth D., Farquhar G. and Griffiths H., et al. (2016) A global dataset of leaf delta 13C values. Scientific Data

Reviewer 2 ·

Basic reporting

fine

Experimental design

fine

Validity of the findings

fine

Additional comments

This manuscript provides novel data on the carbon and nitrogen isotope compositions of plants in Elk Island National Park, Canada. There are not many studies that provide raw CN isotopic data for plants in such ecozones, so the manuscript is quite welcome from that standpoint. The main insights are not really novel, but I don’t see that as a problem. This is simple baseline work that many are too lazy to do before archaeological application, so score one for the author there. The sample sizes are small for a study of this nature, especially for the November sampling period. This is not something that can be remedied, but I don’t think it is fatal. It might be wise to acknowledge this, though, especially as it can make comparisons between various data subsets a bit problematic, and might have relevance for the discussion of skewness.

I would say the one area where this could really use some revision would be the discussion. For instance, in discussion of the skew in nitrogen isotope data, one can readily look to literature from other parts of the world. And one could dig a bit deeper into the habitat-related nitrogen isotope results. In short, the results are not terribly surprising, especially as the aridity effect is quite subtle--something like 2.6 permil for an order of magnitude change in precipitation. Thus, one might only expect miniscule changes here, and these should be largely undetectable given the variation and sample sizes. Much more likely to be driving things would be things like changes in nitrogen deposition by herbivores etc. and wetland related issues that abound in the literature (and yes, this complex and sometimes contradictory).

Anyway, my suggestion would be to take a good day to flesh out the discussion, and that would include using literature from other parts of the world when it comes to things like habitat or plant part based differences. C4 biomes are perfectly acceptable comparators in the discussion, and this would make the contribution more broadly useful.

·

Basic reporting

Metcalfe’s manuscript is well-written and provides good background information, but there are a few areas where some additional background and context are necessary.

Lines 48-59: I know this is just a summary of factors that may lead to variation in the carbon values of C3 plants, but the canopy-effect should be described here since it is offered as a potential explanation of results in the discussion later on.

Lines 74-75: This is a rather minor comment, but it is mentioned in a number of places that having plant isotopic baselines would be helpful to archaeologists/paleontologists. I do not disagree with this point, but a sentence or two specifically demonstrating the applicability would be helpful. A counter to the claim that these studies are important for researchers who study past ecosystems/animal diets would be that the plants occupying these regions of the world in the past were different and/or that the climatic and environmental regimes were different, so how do modern baselines or databases of plant values help archaeologists/paleontologists? Is it general principles – such as archaeologists/paleontologists should be aware that different parts of C3 plants might have different values? Is there value to archaeologists/paleontologists in knowing the specific baseline values?

Table 1: This may be a journal formatting issue, but the inclusion of full citations as a long footnote to the table seems a bit odd. Perhaps include a quick in-text like reference – for example, Zhou et al. 2016 in the ‘Ref.’ column – and integrate the full citations into the overall reference list.

Experimental design

Research question is clearly defined and both analytical and statistical methods are described in sufficient detail. The author provides information on research permits and the permit number is provided in the text.

There are two areas where components of the research design/methods could be clarified.

Lines 129-136: It would be helpful to have some of the numbers of whole-plant samples versus flower/seed samples provided here. Alternatively, an additional table showing the breakdown of the 165 total samples into how many from each habitat in each season, as well as whole samples versus part samples may be helpful to the reader. While this can be reconstructed from Table 2, it takes the reader quite a bit of time to figure out all of the sample numbers.

Lines 144-145: Where the parameters for excluding samples/values is described, the cut-off value for nitrogen gas peaks seems appropriate, but it is unclear whether this means that only nitrogen values were excluded or if both nitrogen and carbon values were excluded for those samples. From Table 2, there are 24 samples without nitrogen values. Are these the values that are excluded, or were there additional samples that did not make it into the final study? Also – what constituted ‘poor reproducibility’?

Validity of the findings

Some minor improvements can be made to the reporting of results and the conclusions.

In the Results section, a summary statistics table may be helpful with means, standard deviations, and medians reported for the various analytical groups (total samples, habitats, seasons, etc.). Some of these are provided in separate tables (Tables 3 and 4), but not all summary statistics are readily available. For instance, on Line 183 and Line 196 reference is made to median values, but the values themselves are not provided. Instead, reference to box-plot figures are provided from which medians can be deciphered/estimated, but it would be much easier on the reader to have these values listed in a table.

Line 182-183: With only 3 samples of horsetail I’m not sure that the statistically significant difference found with forbs is reliable. I am also not sure how such a small sample group would affect the overall ANOVA (for carbon) and Kruskal-Wallis (for nitrogen) tests. Perhaps re-run these without horsetail to see if there is a difference.

Throughout the results, p-values are sometimes given as negative exponents. This is only appropriate for those results with Bonferroni correction; otherwise, p-values reported as negative exponents do not provide useful information. Instead, p-values less than .001 should be reported as p<.001.

Lines 235-246: The finding that nitrogen values are skewed is certainly interesting, but is it possible that this is an artifact of the un-balanced collection of samples by habitat? In Lines 287-291 it is mentioned that nitrogen values are expected to be lower in wet versus dry environments (although, the finding here is in the opposite direction of that expected). Since only 10 nitrogen samples come from wet habitats, is the identification of the skew valid? I do not have an answer to this, but it may be worthwhile to provide other potential reasons for the skew or further justify that the skew is real.

Lines 310-312: It is clear that there are seasonal differences in plant carbon and nitrogen values, and that there are differences in values between plant parts, but at the level of 1-2‰ it is difficult to see how researchers, either archaeologists/paleontologists studying the past or even modern ecologists, could use this difference to infer seasonal patterns from values of animal tissues. When diet-tissue fractionation and individual physiological variation is figured in – will researchers be able to make any claims about seasonal differences or selective feeding on different plant parts (i.e. Lines 280-283)? Wouldn’t these differences just be swamped out unless one had incredibly large sample sizes (say in the hundreds per taxa)?

Additional comments

Metcalfe’s manuscript is an important contribution filling in a major gap in carbon and nitrogen isotopic baselines for C3 plants. In addition to the formal review sections, I have a few minor in-text comments that I will include here.

Abstract – Lines 21-22: I suggest here and elsewhere using ‘bulk’ to distinguish mixed-part samples from specific part (flower/seed) samples.

Lines 34-35: While it is clear what this sentence is getting at, it is awkwardly worded. In particular, the phrase ‘is the foundations’ as the ‘foundation’ being referred to seem to be the ‘significant difference.’ Perhaps just a need to clean up singular versus plural forms.

Line 41: This sentence reads as if herbivores and carnivores occupied different isotopic niches when I think the point is that different isotopic niches can be discerned between both groups of herbivores and groups of carnivores. The former would be expected – herbivores and carnivores should have different isotopic niches. Additionally, as this is a study about plant values, the inclusion of carnivores to the discussion requires additional explanation of how trophic-level plays into carbon and nitrogen values. Since this is not crucial to the current study, I may suggest just removing reference to carnivores.

Line 87-88: Is the motivation to understand bison dietary-selectivity in the present or the past? Or both? This is where the application to archaeology/paleontology versus ecology of the present study is a bit unclear.

Line 107: I think the climatic/environmental information for the Elk Island National Park should have a citation associated with it.

Lines 109-111: The distribution of wood versus plains bison is a bit unclear. In Lines 85-87 it is stated that, historically, wood bison territory was in the north and plains bison territory was in the south, but here plains bison are in the northern part of the park and wood bison in the southern part. Why the switch? When did this happen?

Line 229: Which other studies have found carbon values to be normally distributed? Citations would be helpful here.

---

## Round 0.2 · Minor Revisions

Dear Dr Metcalfe,

Thank you for your revised manuscript. I appreciate the attention to detail and am very satisfied with your responses.

Please can you deal with a few of my very minor comments in the attached file. There are a few remaining points (to assist the reader) that I would appreciate it if you could review. These should not take up much of your time and I look forward to receiving your revised manuscript shortly.

Kind regards,
Alastair Potts

---

## Round 0.3 · accepted · Accept

Thanks for dealing with those last points. I hope you will consider PeerJ for your next paper on Bison hair.